# Evolutionary history of the Arctic flora

Jun Zhang [1,2,3,10], Xiao-Qian Li [1,3,4,10], Huan-Wen Peng[1,3,4], Lisi Hai[1,3,4], Andrey S. Erst [5], Florian Jabbour [6], Rosa del C. Ortiz[7], Fu-Cai Xia [2] ✉, Pamela S. Soltis [8] ✉, Douglas E. Soltis [8,9] ✉ & Wei Wang [1,3,4] ✉

The Arctic tundra is a relatively young and new type of biome and is especially sensitive to the impacts of global warming. However, little is known about how the Arctic flora was shaped over time. Here we investigate the origin and evolutionary dynamics of the Arctic flora by sampling 32 angiosperm clades that together encompass 3626 species. We show that dispersal into the Arctic and in situ diversification within the Arctic have similar trends through time, initiating at approximately 10–9 Ma, increasing sharply around 2.6 Ma, and peaking around 1.0–0.7 Ma. Additionally, we discover the existence of a long-term dispersal corridor between the Arctic and western North America. Our results suggest that the initiation and diversification of the Arctic flora might have been jointly driven by progressive landscape and climate changes and sea-level fluctuations since the early Late Miocene. These findings have important conservation implications given rapidly changing climate conditions in the Arctic.

The Arctic covers an area of ~7.11 million km$^2$, around 5% of the Earth's terrestrial surface[1], and plays an essential role in the global climate system as it functions as a large carbon and methane pool with a slow turnover rate[2,3]. This region has been warming at rates 3–4 times the global average over the last 50 years[4,5]. The Arctic tundra, to the north of the natural tree line, is especially sensitive to the impacts of global warming and is of protection concern[6,7]. This biome harbors distinctive biotas that can tolerate harsh environmental conditions associated with a short growing season, low average annual temperature, and extreme seasonality[8,9]. Altogether, about 2130 angiosperm species and subspecies occur in the Arctic and 84 species are endemic to this region (Fig. 1)[10,11]. The composition, density, and distribution of Arctic vegetation have been changing as a result of climate warming[6,12]. Thus, it is urgent to better understand the assembly history of the Arctic tundra.

The origin and evolutionary dynamics of this unique and fragile biota, however, remain poorly understood[13,14]. The present-day tree-less Arctic tundra zone was covered by broad-leaved evergreen forest in the latest Paleocene-Early Eocene (c. 50–56 Ma), and then by dense coniferous forest in the cool intervals of the Eocene[15,16]. Paleoenvironmental reconstructions support a warm and ice-free environment for the Arctic during the middle Eocene (c. 45 Ma)[17]. Since the Eocene/Oligocene transition (c. 34 Ma), and especially the middle Miocene Climatic Optimum (c. 17–14 Ma), global climate deteriorated drastically[18] and latitudinal temperature gradients increased sharply[19]. Macrofossil evidence suggests that the Arctic tundra developed at the end of the Neogene or in the earliest Pleistocene (c. 3–2 Ma)[16,20–22], in agreement with the time when global temperatures decreased sharply[18]. However, molecular phylogenetic analyses indicate that the ancestors of some Arctic lineages originated in the mid- to late Miocene[13,23–25], though these evolutionary studies have mostly been limited to single taxa. To determine the actual timeframe during which modern Arctic flora began to appear, a molecular phylogenetic study of multiple clades across the angiosperm tree of life is required[26,27].

Immigration, in situ speciation, and local extinction are the mechanisms that form local communities of species. Molecular

[1]State Key Laboratory of Systematic and Evolutionary Botany, Institute of Botany, Chinese Academy of Sciences, 100093 Beijing, China. [2]Forestry College, Beihua University, 132013 Jilin, China. [3]China National Botanical Garden, 100093 Beijing, China. [4]University of Chinese Academy of Sciences, 100049 Beijing, China. [5]Central Siberian Botanical Garden, Russian Academy of Sciences, Zolotodolinskaya str. 101, Novosibirsk 630090, Russia. [6]Institut de Systématique, Evolution, Biodiversité (ISYEB), Muséum national d'Histoire naturelle, CNRS, Sorbonne Université, EPHE, Université des Antilles, Paris 75005, France. [7]Missouri Botanical Garden, 4344 Shaw Blvd, St. Louis, MO 63110, USA. [8]Florida Museum of Natural History, University of Florida, Gainesville, FL 32611, USA. [9]Department of Biology, University of Florida, Gainesville, FL 32611, USA. [10]These authors contributed equally: Jun Zhang, Xiao-Qian Li. ✉e-mail: xfc0707@163.com; psoltis@flmnh.ufl.edu; dsoltis@ufl.edu; wangwei1127@ibcas.ac.cn

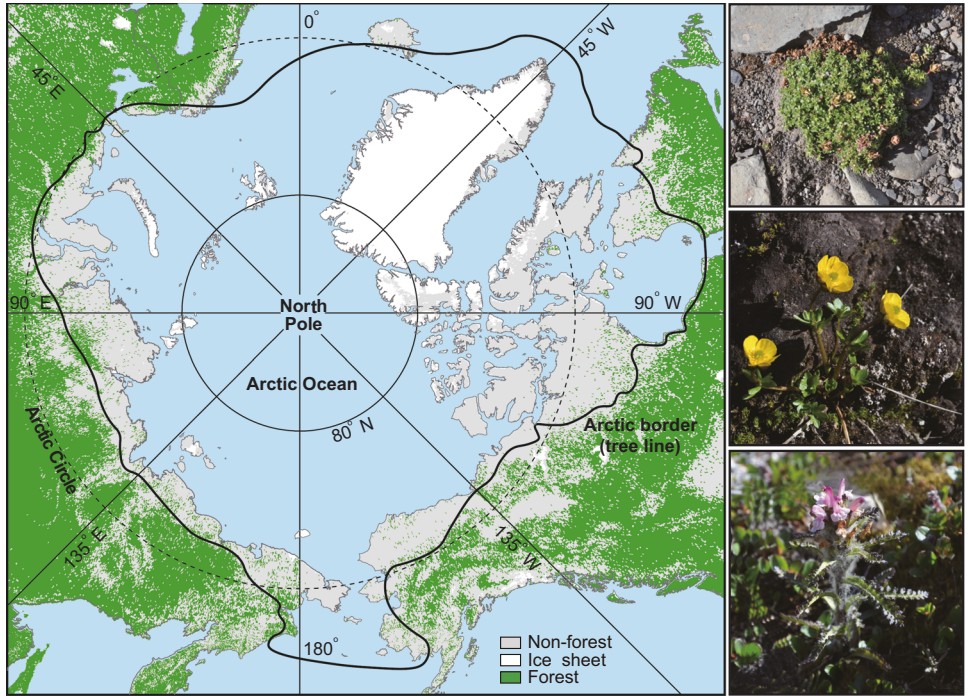

**Fig. 1 | Geographic map of the Arctic.** The range of the Arctic (black line) is based on Elven et al. [11]. Representative plants of the Arctic are shown on the right (from the upper: *Saxifraga oppositifolia, Ranunculus nivalis,* and *Pedicularis hirsuta*).

phylogenetic and phylogeographic studies suggest that Arctic plants are mainly derived from ancestral lineages that occurred in high mountains and adjacent areas to the south of the Arctic[24,25,28–30]. However, most studies focusing on source areas of Arctic species lack divergence time estimation and/or model-based ancestral range reconstruction[13,24,25,28,29]. To date, we know little about the colonization dynamics of the Arctic through time. In addition to immigration, there is evidence that some in situ speciation events have contributed to the formation of the Arctic flora[13,31,32]. Whether both dispersal into the Arctic and Arctic in situ speciation have similar trends through time remains unknown, however, due to a lack of concerted analyses of numerous exemplar Arctic taxa.

We used a multi-taxon analysis to investigate how the Arctic flora was shaped over time. We attempt to determine the initiation age of the Arctic flora and to test whether dispersal and in situ speciation through time have similar trends. Moreover, as it is known that biotic evolution is highly correlated with geological and climatological changes[33], we also explore the potential environmental factors underlying the formation of the Arctic flora.

To unravel the evolutionary history of the Arctic flora, we selected 32 clades and 3626 species, of which 548 are distributed in the Arctic and 40 are restricted to the Arctic. Although it offers the most comprehensive evolutionary analysis of the Arctic tundra to date, this study only covers about 26% of Arctic angiosperm diversity. Nonetheless, we expect that the sampled taxa are representative of the Arctic flora as a whole, as our selected taxa consist of species with various habit and life history traits and belong to 10 orders and 16 families across the angiosperm tree of life (Supplementary Fig. 1; Supplementary Table 1). We reconstructed a time-calibrated phylogeny for each of these 32 clades (Supplementary Figs. 2–27). We then compiled credibility intervals of dispersal and in situ diversification times and calculated their maximal number of observed diversification events (MDEs), respectively. To determine whether the immigrants of the Arctic originated from pre-adapted lineages or were subjected to habitat shifts, we also inferred the ancestral habitat states for each clade for the Arctic endemic species.

In this work, we show that both long-term dispersal and in situ speciation may have contributed to Arctic flora assembly, in association with landscape and climate changes and sea-level fluctuations since the early Late Miocene. We discover that the Arctic immigrants consist of pre-adapted lineages to cold and open habitats and that western North America is an important source area for Arctic species diversity.

## Results and discussion

Our molecular dating and biogeographic analyses identified 131 biogeographic events related to the Arctic species examined here, including 105 dispersal events and 26 in situ diversification events (Supplementary Data 1). Dispersal events predominated over in situ diversification events by approximately 4 times (105/26), implying that immigrants contributed more than the indigenous elements to the current biodiversity of the Arctic biota[13]. The MDE analyses suggest similar trends of dispersal and in situ diversification through time (Fig. 2a, Supplementary Figs. 2–35). Both dispersal and in situ diversification events began in the early Late Miocene (~10–9 Ma) with a relatively low rate, increased rapidly at about 2.6 Ma, and peaked around 1.0–0.7 Ma (Fig. 2a; Table 1). From each MDE curve, we detected three change points (Supplementary Fig. 36; Table 1, Supplementary Table 2), supporting a stepwise model for the formation of the Arctic flora.

Dispersal into the Arctic began at about 10.2 Ma (95% CI: 10.5–9.1), with *Artemisia* from the Mediterranean and *Pleuropogon* from western North American (Supplementary Figs. 31, 34; Table 1, Supplementary Data 1). Arctic in situ diversification was initiated slightly later, around 9.2 Ma (95% CI: 10.1–6.0), mainly by species of *Artemisia, Puccinellia*, and *Ranunculus* (Supplementary Figs. 29, 31; Table 1, Supplementary Data 1). These suggest that the Arctic flora might have emerged in the early Late Miocene, which is much earlier than the prevailing view that this flora began to appear at about 2–3 Ma[16,20]. During the early Late Miocene (~11 Ma), tectonic activities in Greenland and margins of Eurasia occurred (Fig. 2b; Supplementary Table 3). Together with uplift, river erosion changed the Arctic land surface[34]. A steep and

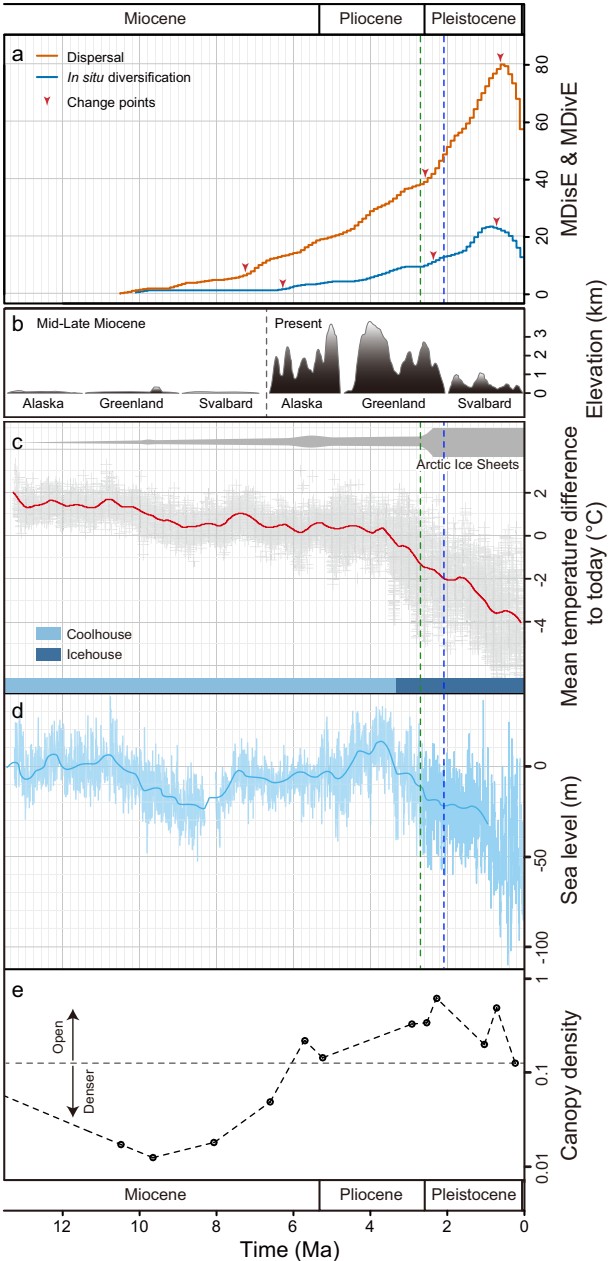

**Fig. 2 | The assembly dynamics of the Arctic flora and its potential driving factors. a** The rates of dispersal and in situ diversification of the Arctic lineages through time. MDisE maximal number of observed dispersal events per Ma. MDivE maximal number of observed in situ diversification events per Ma. **b** Schematic representation of the topography of parts of the Arctic and its adjacent regions in two phases, from the middle Miocene to the present (see Supplementary Table 3 for details). **c** Global temperature inferred from $\delta^{18}O$ levels in benthic foraminifera[18]. **d** Sea-level fluctuations reconstructed from Pacific benthic foraminiferal $\delta^{18}O$ and Mg/Ca records[36]. **e** Changes in forest canopy openness inferred from $C_{est}$ [(shrubs plus herbs)/trees] in northwestern Canada and Alaska[37]. Green and blue dashed lines show a major deep-sea cooling and the first 'deep' glaciation, respectively[40]. Source data are provided as a Source Data file.

steady decline in global temperatures also occurred during this period (Fig. 2c)[18]. Paleoenvironmental reconstruction suggests that the average annual air temperature in the Arctic decreased sharply from 11 °C to 4 °C between 13 and 12 Ma[35]. Also, a marked sea-level drop occurred during the same period (Fig. 2d)[36]. These factors might have led to the formation of new ecological opportunities for cold-adapted lineages and thereby promoted the initiation of the Arctic flora. Pollen and

**Table 1 | The ages of different features of the diversification dynamics**

| Features of the diversification dynamics | Observations (95% confidence interval) (Ma) |
| --- | --- |
| Origination of MDisE | 10.24 (10.50–9.10) |
| Origination of MDivE | 9.20 (10.10–6.00) |
| Peak of MDisE | 0.73 (0.90–0.60) |
| Peak of MDivE | 1.0 (1.10–0.60) |
| Change points of MDisE | 7.23 (7.34–7.12), 2.56 (2.61–2.51), 0.63 (0.65–0.61) |
| Change points of MDivE | 6.25 (6.38–6.12), 2.34 (2.40–2.28), 0.68 (0.70–0.66) |

MDisE maximal number of observed dispersal events per Ma. MDivE maximal number of observed in situ diversification events per Ma.

spore assemblages in northwestern Canada and Alaska indicate that canopy openness began increasing at 9.7 Ma (Fig. 2e)[37].

From 7 to 6 Ma, the rates of both dispersal and in situ diversification started to increase in the Arctic (Fig. 2a; Table 1), temporally in line with the uplift of the Alaska Range and Greenland (Supplementary Table 3). During this period, Northern Hemisphere glaciation was initiated[38,39]. Accordingly, large areas of the continents experienced drying, enhanced seasonality, and a restructuring of terrestrial ecosystems[38]. All of these changes would have facilitated diversification of Arctic lineages. The palynological evidence shows that after 7 Ma herbs and shrubs became important elements in the Arctic flora, and vegetation openness was close to modern levels (Fig. 2e)[37].

Both MDE curves (especially MDisE) increased sharply between 2.6 and 0.6 Ma, peaking at about 0.7 Ma (MDisE) and 1.0 Ma (MDivE) (Fig. 2a; Table 1). The MDE curves of dispersal into the Arctic from western North America, Europe, and Asia display similar patterns (Fig. 3, Supplementary Figs. 37–41). The mean ages of the sampled Arctic species and sampled Arctic endemic species were about 2.66 Ma and 1.60 Ma, respectively (Supplementary Data 2). This timing coincides with a shift to the Icehouse state with global progressive cooling (Fig. 2c)[18,40], as well as with the onset of the large-scale glaciation in the Northern Hemisphere and the establishment of the present climatic regime[40–42]. Owing to the gradual increase of Earth's orbital obliquity[43], seasonality in high northern latitudes increased significantly at 2.6 Ma[42]. Furthermore, the uplift of the Arctic regions, related to glacial isostatic rebound and the dynamics of the ice with variable thicknesses and extents, and erosion from rivers and glaciers resulted in more complex topographic relief (Fig. 2b; Supplementary Table 3)[34,44]. From 2.5 Ma to roughly 1 Ma, high frequent oscillations in sea-level[36], as well as ice sheets repeatedly growing to continental sizes over North America, Greenland and North-West Eurasia[44], might have further repeatedly modified subaerial land. These changes would have promoted the differentiation of microhabitats, increasing the potential for local adaptation, isolation, and lineage diversification. Thus, we propose that the rapid diversification of the Arctic flora was in conjunction with the Pliocene-Pleistocene landscape changes, climatic shifts, and frequent sea-level fluctuations.

From about 0.7 Ma onwards, both MDE curves decreased (Fig. 2a; Table 1), temporally in agreement with extreme climate cooling and an apparent c.100 kyr pacing of glacial and interglacial extremes[45,46]. During the mid-Pleistocene transition (c. 1 to 0.8 Ma), large ice sheets began to display stronger quasi-100-ka cyclicity[36]. These extreme environmental conditions could have prevented immigrants from occupying the Arctic and might have also promoted increased extinction of Arctic species.

Habitat ancestral reconstructions show that the Arctic immigrants whose ancestral habitat type can be inferred unambiguously,

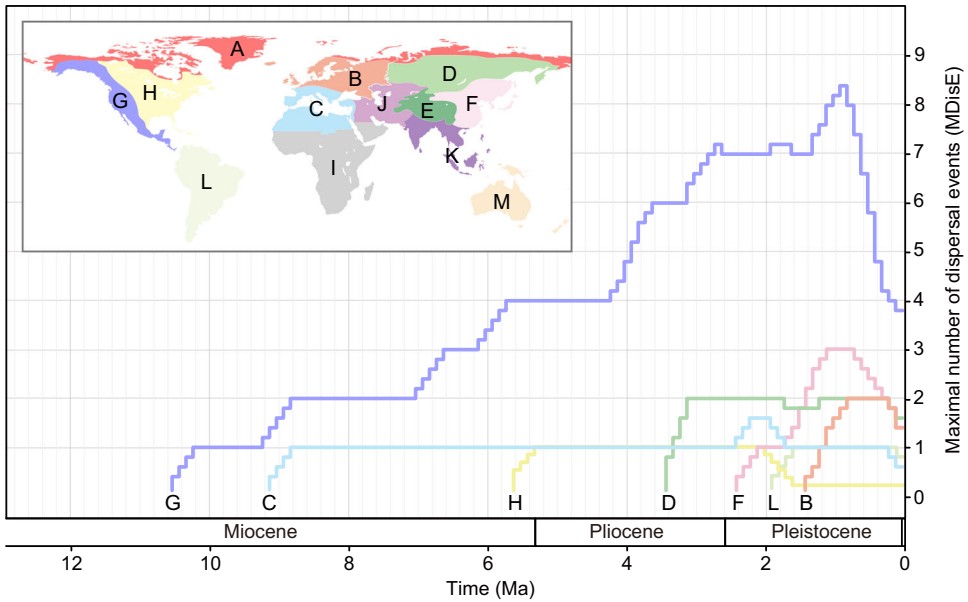

**Fig. 3 | MDisEs from different regions into the Arctic.** A: Arctic, B: North Europe, C: Mediterranean region, D: North Asia, E: Qinghai-Tibet Plateau, F: East Asia, G: western North America, H: eastern North America, I: Africa, J: Turkey-Iran Plateau, K: Southeast Asia, L: South America, M: Oceania. Source data are provided as a Source Data file.

especially endemic species/lineages, are all derived from ancestors inhabiting open habitats in southern high mountains and adjacent lowlands to the south of the Arctic (Supplementary Figs. 42–46). This finding suggests that there was not substantial habitat shift, i.e., the immigrants of the Arctic were pre-adapted to open and cold habitats[14,29]. Our biogeographic analyses with 13 predefined regions show that Arctic species are derived from multiple sources, in agreement with previous results[28,29]. Most biotic dispersals of the Arctic flora were from western North America (~54%, 14/26 of all dispersal events), whereas there were far fewer dispersal events from other regions (Supplementary Table 4). The Arctic interchanges with western North America existed during the entire assembly process of the Arctic flora (Fig. 3), suggesting the possible existence of a long-term dispersal corridor between these two areas. Arctic tundra covers a larger area in North America than in Eurasia (Fig. 1), and the Canadian Arctic Archipelago consists of more than 36,500 islands, a feature not seen in Eurasia. These factors suggest that compared to Eurasia, North American Arctic regions can provide more ecological niches for immigrants of the Arctic.

Our multi-taxon study provides insights into the initiation and assembly dynamics of the Arctic flora over a period of more than 10 million years. Both long-term dispersal and in situ speciation have contributed to the assembly of the Arctic flora, in association with landscape and climate changes and sea-level fluctuations. Our findings will have important conservation implications for the unique and fragile Arctic flora and pinpoint as a conservation priority the dispersal corridor suggested here between the Arctic and western North America.

## Methods
### Data collection
We selected 32 angiosperm clades that contain Arctic and non-Arctic species and have sufficient molecular data available to infer a time-calibrated phylogeny covering the major taxonomic and geographical diversity of each clade (see details in Supplementary Note). We included 3626 species, of which 548 are distributed in the Arctic and 40 are endemic (restricted to the Arctic). These selected taxa belong to 10 orders and 16 families (following APG IV[47]) across the angiosperm tree of life (Supplementary Fig. 1; Supplementary

Table 1). The DNA datasets for the phylogenetic analysis were assembled using sequences that were generated in this study or downloaded from GenBank (Supplementary Data 3). Total genomic DNA was extracted from silica-gel-dried leaves or herbarium specimens using DNeasy Mini Plant Kits (Tiangen Biotech, Beijing, China). The primers used in this study are listed in Supplementary Table 5. PCR products were purified using the Tian quick Midi Purification Kit (TianGen Biotech), and then sequenced using the Bigdye Terminator Cycle Sequencing Kit (Applied Biosystems, ABI) in ABI Prism 3730xl DNA sequencers.

### Phylogenetic analysis and divergence time estimation
For each clade, we performed sequence alignments for each locus in MAFFT v.7.037[48] and manually adjusted with Geneious v.9.1.4[49]. Referring to Li et al. [50], we first conducted maximum likelihood analysis in IQ-TREE v.2.1.2[51] and removed the strongly conflicting taxa (>70% bootstrap values for different placements) between the plastid and nuclear trees[52]. After removing sixty-eight species, all of which are non-Arctic (Supplementary Data 3), a total of 3588 species was used for subsequent analyses. For the final concatenated datasets, we used BEAST v.1.8.4[53] to infer time-calibrated phylogenies using an uncorrelated relaxed molecular clock model, a Yule prior, and the GTR + I + Γ model for each locus separately. All analyses were run for 50 million generations, sampling every 5000 generations. Tracer v.1.7.2[54] was used to assess appropriate burn-in and adequate effective sample size values (ESS > 200). We used fossils or secondary ages as calibration points, depending on the information available for each clade (see details in Supplementary Note).

### Biogeographic and habitat analyses
Ancestral ranges were estimated for each clade in BioGeoBEARS[55] under the dispersal-extinction-cladogenesis model[56]. To minimize the uncertainties associated with phylogenetic inference, we randomly sampled 1000 posterior trees subsampled from the BEAST analysis as a "trees file" and used the maximum clade credibility (MCC) tree as a final representative tree. To infer the differences between the dynamics of dispersal-associated diversification versus in situ speciation of the Arctic species, we classified species into either Arctic (A; north of the natural tree line), non-Arctic (B), or occurring in both

regions (AB), based on their current known distribution. To examine the possible sources of the Arctic species, we also coded 13 biogeographic regions based on the floristic division of Takhtajan[57]: (A) Arctic, (B) North Europe, (C) Mediterranean region, (D) North Asia, (E) Qinghai-Tibet Plateau, (F) East Asia, (G) western North America, (H) eastern North America, (I) Africa, (J) Turkey-Iran Plateau; (K) Southeast Asia, (L) South America, and (M) Oceania (Fig. 3). For each clade, both biogeographic analyses were implemented in RASP 4.2[58]. The maximum number of areas in the ancestral ranges was set as the maximum number of areas observed in the extant taxa, and dispersal probabilities between pairs of areas were specified for two separate time slices (Supplementary Table 6).

Ancestral habitats were reconstructed for each clade with endemic Arctic species using the Bayesian binary MCMC (BBM) method in RASP 4.2[58]. Species were scored as open habitat (without a canopy of other plants; e.g., tundra, grassland, alpine), closed habitat (forest understory with at least a partial canopy), or occurring in both habitats. To take into account phylogenetic uncertainty, the MCC tree and 1000 posterior trees subsampled from BEAST analysis were imported as input files for each clade. All analyses were run for $5\times10^6$ generations under the fixed JC + G model, with 10 simultaneous chains, sampling every 1000 generations.

### Assembly dynamic multi-taxon analysis

Based on the results of the biogeographic analysis with two redefined areas (Arctic and Non-Arctic), we categorized the species-level biogeographic events for the arctic species on the phylogenetic tree (Supplementary Fig. 28) into two types, as suggested in Xu et al.[59], "dispersal event into the Arctic" and "in situ diversification event". To infer general patterns of these two events, we estimated the maximal number of observed dispersal events per Ma (MDisE)[60] and in situ diversification events per Ma (MDivE)[59], respectively. These MDE rates were calculated by summing potential dispersal or in situ diversification events over all data sets through time using time slices of 0.1 million years (based on the divergence time credibility intervals given in Supplementary Data 1). The data were smoothed by calculating mean values using a sliding window approach with a time frame of 0.5 Ma. In addition, we extracted dispersal times from different regions into the Arctic and calculated their respective MDisEs using the same approach.

To assess the 95% confidence intervals of the MDE origination and peaks, we re-sampled 1000 bootstrap pseudoreplicates from the raw credibility intervals (used for inferring the MDisE and MDivE rates above) using the R package *sample* function, and then repeated the calculations to find the MDE origination and peaks. We identified the change points in the MDE curves using the segmented regression method[61] via the R package *segmented* function[62]. We modeled the trend with 1, 2, 3 and 4 change points and used the Bayesian Information Criterion (BIC) to determine the optimal model. The 95% confidence intervals of the change points were calculated using the R package *lines.segmented* function.

### Reporting summary

Further information on research design is available in the Nature Portfolio Reporting Summary linked to this article.

## Data availability

Newly obtained sequences have been deposited in GenBank, the accession numbers are provided in Supplementary Data 3, and vouchers are deposited in Herbarium, Institute of Botany, the Chinese Academy of Sciences, Beijing (PE). Alignments and timetrees for the 32 clades investigated are available at Zenodo (https://doi.org/10.5281/zenodo.7868007) and FigShare (https://doi.org/10.6084/m9.figshare.23506542). Source Data are provided in the Source Data file. Source data are provided with this paper.

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

## Acknowledgements

This research was partially funded by the Strategic Priority Research Program of the Chinese Academy of Sciences (XDB31030000) to W.W., the National Natural Science Foundation of China (32011530072, 32170210 and 31770231) to W.W., K.C. Wong Education Foundation (GJTD-2020-05) to W.W., P.S.S. and D.E.S., and the state assignments performed by the CSBG SB RAS (AAAA-A21-121011290024-5) to A.S.E.

## Author contributions

W.W., F.-C.X, P.S.S., and D.E.S. conceived the research. J.Z., X.-Q.L., H.-W.P., L.H., and W.W. analyzed data. J.Z., X.-Q.L., A.S.E, F.J., R.C.O., P.S.S., D.E.S., and W.W. wrote the paper with inputs from all the authors.

## Competing interests

The authors declare no competing interests.
