## [Peer Review File · Nature Communications]

Reviewers' Comments:

Reviewer #1:

Remarks to the Author:

In this paper, dated phylogenies are generated to determine when species that occur in the present-day arctic flora dispersed into the Arctic or originated there through in situ speciation. In addition, analyses are conducted to determine from which geographical region and from what type of habitat (open or closed) dispersal occurred. It is shown that dispersal of ancestral arctic lineages into the Arctic began in the early Late-Miocene but involved relatively few of such lineages. Dispersal of arctic plant species into the Arctic increased considerably after ~7 Ma and again after ~2.6 Ma, peaking at ~0.8 Ma, with changes in rates of in situ speciation showing a similar historical trend. It was further indicated that Northwest America was a major corridor of dispersal into the Arctic and that arctic species were dispersed from open habitats indicating they were preadapted to the Arctic tundra biome. The strength of the paper is that the findings are based on an analysis of 32 clades that include 517 species distributed in the Arctic of which 54 are endemic to the region. It is pointed out that this covers about a quarter of arctic angiosperm diversity and represents by far the most comprehensive analysis of the evolution of the arctic flora to date. I enjoyed reading this manuscript and view the findings as extremely useful in providing an improved understanding of the evolution of the present-day arctic flora. However, though there is much to admire about the research and findings reported, which I am sure will be highly cited, I do have concerns about how the findings of previous relevant studies have been acknowledged or not acknowledged, and the effects and legitimacy of removing taxa from clades, so as to obtain phylogenetic consistency over loci. I am also surprised that there is no discussion of factors that might favour the dispersal of lineages into the Arctic (geographical and biological), or mechanisms of in situ speciation. Regarding the latter, it has been proposed by others that reticulative polyploidy was important. These and other points that need addressing are listed below and, together with minor edits, are also shown in the marked-up main text and supplementary information files attached.

Main Text

Lines 54-55. The reference (13) given in support refers to a paper published 20 years ago, which mainly focused on locations of refugia for arctic plants during the Quaternary, long distance dispersal, and in situ speciation. True, it was pointed out in this paper that we were largely ignorant about the origins of the 'founding stocks' of the arctic flora, since when several papers have been published reporting dated molecular phylogenies of genera that include arctic taxa to help fill the gap highlighted by Abbott & Brochmann. So, the origin of the arctic flora is much better known now than it was 20 years ago. This should be acknowledged.

62-63. I have not been able to access reference (15). However, I recommend that the key references of Matthews & Ovenden (1990) and Bennike & Bocher (1990) should be cited.

64-66. This is not exactly correct. Previous molecular phylogenetic studies on *Artemisia*, *Carex*, *Cassiope* and *Ranunculus* indicate that ancestral arctic lineages originated in the mid- to late-Miocene (e.g. references 19-23 cited in current paper, and Hou et al. (2016)). The results of Zhang et al. confirm these previous findings for a wider group of clades.

67-69. This statement does not tell the whole story and consequently is too simplistic. For example, it ignores the findings of Hou et al. (2016) which suggest *Cassiope* originated in the Arctic-Boreal region (Beringia) and that species found in high mountains to the south (Himalayas and Hengduan Mts) originated following north-south colonization. See also Sun (2002) who proposed N-S colonization within several other genera containing arctic species.

71. Not entirely accurate. See Brochmann & Brysting (2008) who reviewed studies of in situ speciation in the Arctic. Should not be ignored. See also Brochmann & Alsos (2021).

85. Note that figures are not numbered in Extended Data. Please number and provide legends to figures in Extended Data.

108-110. I think it would be more accurate to say "These suggest that early components of the Arctic flora began to emerge in the early Late Miocene". For many Arctic sensu stricto lineages (in fact the large majority of such lineages, Fig. 2a) dispersal into the Arctic began much later.

118-119. Yes, but did any present-day arctic plants occur in these open habitats? Is there fossil evidence in support of this?

142, 'Eurasia'. Note that ice-sheets did not repeatedly extend to continental sizes over Eurasia. They were limited to North-West Eurasia. North- Central and North-East Eurasia remained largely unglaciated during the Quaternary.

151-152. But see Brochmann & Brysting (2008) who proposed that sub-division of species ranges during glacial periods was a driver of genetic divergence and in situ speciation with the latter sometimes resulting from secondary contact and allopolyploidy during interglacials.

155-156. What about Cassiope (Hou et al, 2016)? Couldn't the ancestral habitat type of this genus be determined?

164-165. Possible reasons for this? Worth speculating that the N-S orientation of the Rockies was possibly a cause of this. But if so, why haven't the Urals acted similarly as a high dispersal corridor?

167-170. Would be informative to mention that polyploidy and cryptic speciation are considered important mechanisms of in situ speciation in the Arctic (Brochmann & Brysting 2008; Gustafsson et al. 2021; Kadereit & Abbott 2021).

296. It is not clear to me what the x-axes for Alaska, Greenland and Svalbard represent in Fig. 2b. Should be made clear.

302-301. Figure 3 needs improving. It is not easy to distinguish clearly some of the curves. Would be helpful to label each curve according to geographical region. Colour alone is not sufficient.

305. Mediterranean region appears to include the Alps and Pyrenees. Were arctic species dispersed from these high mountains, or lowlands as well in this region?

327-328. The number of species removed from each clade should be indicated in Supp. Info. An indication should also be given regarding topology inconsistency across loci and what this might be due to, i.e. incomplete lineage sorting and/or reticulation. Could these causes be distinguished and commented on?

364. Needs explanation. What are the two redefined areas?

Supplementary Information

185. Mandel et al. (2019) is not cited in references.

References:

Bennike, O. & Bocher, J. (1990). Forest-tundra neighboring the North Pole: plant and insect remains from the Plio-Pleistocene Kap Kobenhavn formation, North Greenland. *Arctic* 43: 331–338.
Brochmann C. & Alsos IG. (2021). Origin and dispersal of the North Atlantic vascular plant floras. In

Panagiotakopulu E & Sadler JP (eds) Biogeography in the Sub-Arctic: The past and future of North Atlantic Biota. John Wiley & Sons Ltd. Pp.85-101.

Brochmann, C. & Brysting, A.K. (2008). The Arctic – An evolutionary freezer? *Plant Ecology & Diversity* 1: 181–195.

Gustafsson ALS, Gussarova G, Borgen L, Ikeda H, Antonelli A, Marie-Orleach L, Rieseberg LH, Brochmann C. 2021. Rapid evolution of postzygotic reproductive isolation is widespread in arctic plant lineages. *Ann Bot.* 129:171-184.

Hou Y et al. (2016). RAD-seq data point to a northern origin of the arctic–alpine genus *Cassiope* (Ericaceae). *Mol. Phylo. & Evol.* 95:152-160.

Kadereit, JW. & Abbott R.J. (2021) Plant speciation in the Quaternary. *Plant Ecology & Diversity* 14:105-142.

Sun, H. (2002). Evolution of arctic-tertiary flora in Himalayan–Hengduan Mountains. *Acta Bot. Yunnanica* 24, 671–688.

Reviewer #2:

Remarks to the Author:

The present manuscript is dedicated to the arctic biome from a botanical-evolutionary perspective. It attempts to shed light on the origin and evolutionary dynamics of the arctic flora by examining different phylogenetic lineages of angiospermous plants. It is based on an elaborate evaluation of the phylogenetic trees of several arctic genera. Included were 32 clades with 3,588 species. This makes the study an important stone for understanding this biome, especially in the light of climate change. The term 'clade' is used in a phylogenetic context. It seems that the authors consistently refer to 'genera' in the taxonomic sense (and not families or orders etc.), so I would recommend using this term.

The introductory sections are overall well written and give a brief overview of the geological/climatic/vegetation history of the Arctic. However, most of the references are older than 20 years and more recent studies have not been sufficiently considered. This is also reflected in the statement (lines 54-55): "The evolutionary origin of this unique and fragile biota, however, is poorly understood". This statement was made in 2003! Instead, there has been significant progress on this issue in the last 20 years. Similarly questionable appears the statement (l. 64-65): "Yet, the actual timeframe during which modern arctic flora began to appear remains unknown."

In some other topics, the current state of knowledge is indeed not correctly reflected either. For example, with regard to the relatively minor importance of in situ speciation in the Arctic (l. 71), which has already been demonstrated in previous studies, e.g. for *Draba*, *Douglasia*, *Saxifraga*, *Cerastium*, etc. In this respect, the present study offers no new examples or findings at all.

A study comparable to the present work, entitled "Taxon recruitment of the arctic flora: an analysis of phylogenies" (Hoffmann & Röser, *New Phytologist* 182: 774-780), was published in 2009. In this investigation (p. 774) "available molecular phylogenetic studies were evaluated for 148 of 374 genera occurring in the Arctic to determine the relative roles of their independent origins and their diversification in the development of the contemporary arctic flora". Many of the results of this work anticipate supposedly new findings of the present manuscript. However, it is not cited at all in the present manuscript, which is a rather big mistake. The by far much more important role of dispersal compared to in situ evolution of arctic species (l. 96) additionally is not new.

The discussion of MDEs identifies "three change points ..., supporting a stepwise model for the formation of the arctic flora" (l. 102). Undoubtedly, there have been many changes over the last 3 million years, including interglacials with less glaciation in the Arctic than today, during which the arctic biome, including its former arctic flora, probably almost disappeared. The space available for arctic flora was certainly severely affected. The role of extinction should be investigated in this context and time frame. The explanation for the drop in the MDE curves at about 0.8 Ma is not convincing (lines 148-152).

Conclusions such as "dispersal into the Arctic began around 11.8 Ma" (l. 103) and "dispersal into the Arctic began at about 11.8 Ma" (l. 106) also appear questionable, leading to the assumption "that

arctic flora had emerged in the early Late Miocene, which is much earlier than the prevailing view that this flora began to appear at about 2–3 Ma". The most likely explanation is missing from the manuscript. Some phylogenetic lineages with arctic taxa are older than the arctic biome, as noted already in previous studies.

The conclusion that "most biotic dispersal of the arctic flora has been between the Arctic and western North America, whereas there were far fewer dispersal events from other regions" (lines 160-165) should be carefully reviewed. Dispersal along western North America into the Arctic is undoubtedly important and has also been noted in previous studies, but the connections of the vast Eurasian Arctic in particular with southern regions have probably been underestimated in this study. One possible reason is the insufficient consideration of the Eurasian Arctic in this study. As can be seen from the "supplementary information" on the topic "Geographical range and habitat information were mainly compiled from the following sources", no flora at all from the area of the former USSR or the famous "Arkticheskaya Flora" (Tolmachev, A.I. (Ed.), 1960–1987. Arkticheskaya Flora SSSR. Vol. 1-10. Moscow, Leningrad) was taken into account, which is actually not possible in this matter. The important relations between the Eurasian Arctic and the South Siberian mountains are not mentioned anywhere in the manuscript. See: Tkach, N.V., Röser, M., Hoffmann, M.H. (2008): Range size variation and diversity distribution in the vascular plant flora of the Eurasian Arctic. *Organisms, Diversity and Evolution* 8: 251-266.

The supplement also states that "Phylogenetic data using DNA sequences were generated in this study or collected from GenBank". However, a random check of the sequence numbers in GenBank did not reveal any sequence that was generated in the present study. In the case of *Draba*, for example, it was even explicitly stated that the *matK* and *rbcl* sequences had been determined in the context of this study. This was also not true. It would be practical if the new sequences were labelled.

I checked only some of the genera studied to see if the arctic species were correctly selected (see Panarctic Flora <http://nhm2.uio.no/paf>, distribution maps in 'Arkticheskaya Flora USSR', other sources). I found that several entries were not correct, that some species were not considered as "arctic" or that some arctic species had been overlooked. Endemic arctic species were marked in red, although in many cases these species have a wider distribution range. Unfortunately, the number of such data is quite high, so this should be checked again for all species studied and the analyses should be done again if necessary.

I also checked a few of the genera ("clades") studied in the manuscript to see if the arctic species were correctly marked as such (see Panarctic Flora, distribution maps in Arkticheskaya Flora USSR, other sources). I found that several markings on the phylogenetic trees were incorrect: some species were not marked as arctic or that some arctic species were overlooked. Endemic arctic species were marked in red on the trees, although in many cases these species have a wider distribution range. Unfortunately, the number of such data is quite high, so this should be checked again for all species studied and the analyses should be done again if necessary.

Some examples:

- *Artemisia arctica*, *A. tilesii*, *A. leucophylla*, *A. gmelinii*, *A. frigida* are overlooked as arctic species, whereas *A. borealis* is not endemic arctic.
- *Delphinium elatum* is not endemic arctic.
- *Plantago canescens* is not endemic arctic.
- *Plantago elongata* is not arctic.
- *Braya humilis* and *B. glabella* are not endemic arctic.
- *Braya siliquosa* is arctic species.
- *Oxytropis middendorffii* is not endemic arctic.
- *Ranunculus glacialis* is not endemic arctic.
- *Androsace triflora*, *A. chamaejasme*, *A. septentrionalis*, *A. filiformis* are arctic.
- *Papaver lapponicum*, *P. radicatum*, *P. mcconnellii* are arctic.
- *Papaver nudicaule* is not endemic arctic.
- *Senecio lugens*, *S. pseudoarnica* are arctic.

Please check the manuscript: Arctic with upper case "A" as region vs. arctic (for example, arctic biome or arctic species) with lower case "a" as adjective.

Table S8: Reference (72) seems to be wrong, it should be (71).
Fig. 3: Explanation of "E" is missing.

All in all, a somewhat careless handling of facts can be observed here as well. One cannot avoid the impression that the actual state of current knowledge has not been adequately taken into account in many places in this study and that many things are presented as supposedly new, although they have been known for a long time.

Reviewer #3:
None

Response to Reviewers

Reviewer #1 (Remarks to the Author):

In this paper, dated phylogenies are generated to determine when species that occur in the present-day arctic flora dispersed into the Arctic or originated there through in situ speciation. In addition, analyses are conducted to determine from which geographical region and from what type of habitat (open or closed) dispersal occurred. It is shown that dispersal of ancestral arctic lineages into the Arctic began in the early Late-Miocene but involved relatively few of such lineages. Dispersal of arctic plant species into the Arctic increased considerably after ~7 Ma and again after ~2.6 Ma, peaking at ~0.8 Ma, with changes in rates of in situ speciation showing a similar historical trend. It was further indicated that Northwest America was a major corridor of dispersal into the Arctic and that arctic species were dispersed from open habitats indicating they were preadapted to the Arctic tundra biome. The strength of the paper is that the findings are based on an analysis of 32 clades that include 517 species distributed in the Arctic of which 54 are endemic to the region. It is pointed out that this covers about a quarter of arctic angiosperm diversity and represents by far the most comprehensive analysis of the evolution of the arctic flora to date. I enjoyed reading this manuscript and view the findings as extremely useful in providing an improved understanding of the evolution of the present-day arctic flora. However, though there is much to admire about the research and findings reported, which I am sure will be highly cited, I do have concerns about how the findings of previous relevant studies have been acknowledged or not acknowledged, and the effects and legitimacy of removing taxa from clades, so as to obtain phylogenetic consistency over loci. I am also surprised that there is no discussion of factors that might favour the dispersal of lineages into the Arctic (geographical and biological), or mechanisms of in situ speciation. Regarding the latter, it has been proposed by others that reticulative polyploidy was important. These and other points that need addressing are listed below and, together with minor edits, are also shown in the marked-up main text and supplementary information files attached.

Response 1: Thank you very much for your positive comments. We are pleased to see that you think that "there is much to admire about the research and findings reported, which I am sure will be highly cited".

We sincerely thank you for the references that you listed. In the revised manuscript, we have now acknowledged previous relevant studies (Lines 48–49, 56–60, 67–69, 70–71, 90–101; see details in the below Responses 2, 3 and 6 to Reviewer 1).

It is routine to exclude the strongly conflicting taxa in plastid and nuclear trees before

combining plastid and nuclear datasets (reviewed by Wang et al. 2014 *Sci. China Life Sci.* 57, 280–286). In this study, we selected 32 clades and 3,626 species. For each clade, we first conducted maximum likelihood analysis and removed the strongly conflicting taxa (> 70% bootstrap values for different placements) between the plastid and nuclear trees (Lines 205–209). The species that we removed from subsequent analyses are non-Arctic species (Lines 210–211; see revised Supplementary Table 7 for details). We also stress that in this study our major objective is to investigate the occurrence variations of the Arctic species through time, not to investigate the evolutionary history of each clade that we selected. Please see below Response 18 to Reviewer 1.

Previous studies have indicated that the Arctic species/lineages mainly originated from southern high mountains and adjacent lowlands to the south of the Arctic (e.g., Hoffmann et al. 2017 *Am. J. Bot.* 104, 1334–1343 and references therein; Qian et al. 2022 *Global Ecol. Biogeogr.* 31, 396–404), and that many Arctic plants have extreme long-distant colonization ability (Alsos et al. 2007 *Science* 316, 1606–1609; Brochmann & Brysting, 2008 *Plant Ecol. Divers.* 1, 181–195). In this study, our major objective is to assess the colonization dynamics of the Arctic and is not to examine factors that might favor the dispersal of lineages into the Arctic, which have already been well-reviewed (see Brochmann & Brysting, 2008 *Plant Ecol. Divers.* 1, 181–195). Please see below Response 5 to Reviewer 1.

In this study, our taxon sampling is at the species level. Thus, it is not practical for us to delimit species with polyploidy and cryptic speciation—that would entail another massive investigation. Our major objective is to assess *in situ* speciation dynamics through time across the entire Arctic, as well as colonization dynamics of the Arctic (Lines 74–77) and is not to explore the potential mechanisms of *in situ* diversification, which is beyond the scope of this study. Please see below Response 14 to Reviewer 1).

We sincerely thank you very much for your invaluable suggestions. Using your helpful comments and edits, we made corrections and improvements, and all minor edits that you made on the pdf version have been accepted. We made a point-by-point response to all your below comments as well as to those in the marked-up main text and supplementary information files. We hope that you will be satisfied with our revisions and responses below.

Main Text

Lines 54-55. The reference (13) given in support refers to a paper published 20 years ago, which mainly focused on locations of refugia for arctic plants during the Quaternary, long distance dispersal, and *in situ* speciation. True, it was pointed out in this paper that we were largely ignorant about the origins of the 'founding stocks' of the arctic flora,

since when several papers have been published reporting dated molecular phylogenies of genera that include arctic taxa to help fill the gap highlighted by Abbott & Brochmann. So, the origin of the arctic flora is much better known now than it was 20 years ago. This should be acknowledged.

Response 2: Yes, you are right. Our understanding of the evolutionary origins of the Arctic flora has indeed improved in recent years through molecular phylogenetic analyses, but most studies focused on source areas of Arctic species without divergence time estimation and/or model-based ancestral range reconstruction; moreover, evolutionary studies in the Arctic have mostly been limited to single taxa (see below Response 4 to Reviewer 1). In this study, we used a multi-taxon analysis to investigate how the Arctic flora was shaped over time, which is still poorly understood. In the revised manuscript, we thus first changed "The evolutionary origin of this unique and fragile biota," to "The origin and evolutionary dynamics of this unique and fragile flora," (Line 48). We then deleted the initial Reference 13 and cited Hoffmann & Röser (2009 *New Phytol.* 182, 774–780; Qian et al. 2022 *Global Ecol. Biogeogr.* 31, 396–404) (Line 49). Also please see below Response 2 to Reviewer 2.

62-63. I have not been able to access reference (15). However, I recommend that the key references of Matthews & Ovenden (1990) and Bennike & Bocher (1990) should be cited.

Response 3: The initial Reference (15) is from Jack A. Wolfe, who discussed the initial time of formation of the Arctic tundra by reviewing macrofossil evidence. One can download the initial reference (15) from the private link that we provided: https://datadryad.org/stash/share/IeJiW48uzzd6iHj_VSKq697b298NCVLbd35wx_Ua5-Q. Following your suggestion, we added the citations of Bennike & Bocher (1990 *Arctic* 43, 331–338) and Matthews & Ovenden (1990 *Arctic* 43, 364–392) (Line 57).

64-66. This is not exactly correct. Previous molecular phylogenetic studies on *Artemisia*, *Carex*, *Cassiope* and *Ranunculus* indicate that ancestral arctic lineages originated in the mid- to late-Miocene (e.g. references 19-23 cited in current paper, and Hou et al. (2016)). The results of Zhang et al. confirm these previous findings for a wider group of clades.

Response 4: Fossil and molecular data seem to provide different times of origin for the Arctic flora (Lines 49–58). In our revision, we first added to the introduction a summary of results of molecular phylogenetic studies, "Molecular phylogenetic analyses indicate that the ancestors of some Arctic lineages originated in the mid- to late Miocene. However, these evolutionary studies have mostly been limited to single taxa" (Lines 58–60). The sentence "Yet, the actual timeframe during which modern Arctic flora began to appear remains unknown" was then rephrased to "To determine the actual timeframe during which modern Arctic flora began to appear, a molecular phylogenetic study of

multiple clades across the angiosperm tree of life is vital" (Lines 61–63).

67-69. This statement does not tell the whole story and consequently is too simplistic. For example, it ignores the findings of Hou et al. (2016) which suggest *Cassiope* originated in the Arctic-Boreal region (Beringia) and that species found in high mountains to the south (Himalayas and Hengduan Mts) originated following north-south colonization. See also Sun (2002) who proposed N-S colonization within several other genera containing arctic species.

Response 5: In this study, our major objective is to investigate how the Arctic flora was shaped over time, i.e., colonization dynamics of the Arctic and *in situ* diversification process in the Arctic. The out-of-Arctic migration to other areas is beyond the scope of our study, i.e., it is not necessary to mention N-S colonization here.

71. Not entirely accurate. See Brochmann & Brysting (2008) who reviewed studies of *in situ* speciation in the Arctic. Should not be ignored. See also Brochmann & Alsos (2021).

Response 6: Thank you very much for your reminder. This part has been re-written. First, we added *in situ* speciation in the Arctic and cited these two papers that you suggested (Brochmann & Brysting, 2008 *Plant Ecol. Divers.* 1, 181–195 and Brochmann & Alsos, 2021 in *Biogeography in the Sub-Arctic: The past and future of North Atlantic Biota* 85–101), as well as the paper of Hoffmann & Röser (2009 *New Phytol.* 182, 774–780) (Lines 70–71). Second, the last sentence in the paragraph has been rephrased to "Whether both dispersal into the Arctic and Arctic *in situ* speciation have similar trends through time remains unknown, however, due to a lack of concerted analyses of numerous exemplar arctic taxa" (Lines 71–73).

85. Note that figures are not numbered in Extended Data. Please number and provide legends to figures in Extended Data.

Response 7: Based on the format requirement of *Nature Communications*, we first changed "Extended Data" to "Supplementary" throughout the main text, and then added the figure number in each Supplementary figure. In our initial submission, the legends to figures were provided at the end of the main text. In this revised version, we moved the legends to supplementary figures into the file Supplementary Information (Pages 3–4).

108-110. I think it would be more accurate to say "These suggest that early components of the Arctic flora began to emerge in the early Late Miocene". For many Arctic *sensu stricto* lineages (in fact the large majority of such lineages, Fig. 2a) dispersal into the Arctic began much later.

Response 8: Our meta-analysis indicates that Mediterranean *Artemisia* and western North American *Pleuropogon* dispersed into the Arctic in the early Late Miocene, and during the same period, Arctic *in situ* diversification occurred in three genera (*Artemisia*, *Puccinellia*, and *Ranunculus*) (Lines 108–113). These indeed suggest that some components of modern Arctic flora might have formed during this period. The above-mentioned four genera are not related and distributed in three distantly related families of angiosperms. Thus, we think that it is appropriate to state "Arctic flora might have emerged in the early Late Miocene" (in the revised manuscript, we changed "had emerged" to "might have emerged" to tone down our conclusion; Lines 113–115). A similar case can be found in Ding et al. (2020 *Science* 369, 578–581), who said "Our historical reconstruction indicate that an alpine flora had emerged in the THH region by the early Oligocene, mainly by ancestral lineages of Delphineae and Saxifragaceae".

118-119. Yes, but did any present-day arctic plants occur in these open habitats? Is there fossil evidence in support of this?

Response 9: Yes, there are some present-day Arctic plants found in these two open habitats, such as *Puccinellia borealis*, *Papaver mcconnellii*, and *Smelowskia borealis*. This open habitat was indeed inferred from fossil data, i.e., pollen and spore evidence (White et al. 1997 *Paleogeogr. Paleoclimatol. Paleoecol.* 130, 293–306) (Lines 124–125).

142, 'Eurasia'. Note that ice-sheets did not repeatedly extend to continental sizes over Eurasia. They were limited to North-West Eurasia. North-Central and North-East Eurasia remained largely unglaciated during the Quaternary.

Response 10: Following your suggestion, we have changed "Eurasia" to "North-West Eurasia" (Line 150).

151-152. But see Brochmann & Brysting (2008) who proposed that sub-division of species ranges during glacial periods was a driver of genetic divergence and *in situ* speciation with the latter sometimes resulting from secondary contact and allopolyploidy during interglacials.

Response 11: In this study, we sampled 32 clades and 3,626 species (Lines 80–82), i.e., our taxon sampling is at the species level, not at the subspecies or population level. Thus, it is well beyond the scope of study for us to discuss the effect of the sub-division of species ranges on *in situ* speciation. The same can be found in Ding et al. (2020 *Science* 369, 578–581) and Xu et al. (2021 *Nat. Sci. Rev.* 8, nwa263), who also only explore *in situ* diversification dynamics at the species level. Nonetheless, this sentence has been rephrased "These extreme environmental conditions could have prevented immigrants

from occupying the Arctic and might have also promoted increased extinction of Arctic species" (Lines 160–162; please see below response 5 to Reviewer 2).

155-156. What about *Cassiope* (Hou et al, 2016)? Couldn't the ancestral habitat type of this genus be determined?

Response 12: In this study, we only reconstructed ancestral habitats for those clades with Arctic endemic species (Lines 90–93, 238–239), because it is difficult to determine ancestral habitats for non-Arctic endemic species. In *Cassiope*, three species occur in the Arctic, but all are not restricted to the Arctic. Thus, we did not perform an ancestral habitat reconstruction for this genus.

164-165. Possible reasons for this? Worth speculating that the N-S orientation of the Rockies was possibly a cause of this. But if so, why haven't the Urals acted similarly as a high dispersal corridor?

Response 13: In the revised manuscript, we provide a possible reason for this, "Arctic tundra covers a larger area in North America than in Eurasia (Fig. 1), and the Canadian Arctic Archipelago consists of more than 36,500 islands, a feature not seen in Eurasia. These factors suggest that compared to Eurasia, North American Arctic regions can provide more ecological niches for immigrants of the Arctic" (Lines 175–179).

167-170. Would be informative to mention that polyploidy and cryptic speciation are considered important mechanisms of *in situ* speciation in the Arctic (Brochmann & Brysting 2008; Gustafsson et al. 2021; Kadereit & Abbott 2021).

Response 14: Very good suggestion! However, as we explained in above Response 11 to Reviewer 1, in this study we investigate "evolutionary history of the Arctic flora" at the species level. It is difficult to delimit the species with polyploidy and cryptic speciation. In this study, our major objective is to assess *in situ* speciation dynamics through time in the whole Arctic, as well as colonization dynamics of the Arctic (Lines 74–77), and is not to explore potential mechanisms of *in situ* diversification, which is beyond the scope of this study. Moreover, it is our conclusion "long-term dispersal and *in situ* speciation have contributed to the assembly of the Arctic flora" (Lines 181–183), not discussion, in which you made this comment. Thus, it seems to be inappropriate here to mention the mechanisms of *in situ* speciation.

296. It is not clear to me what the x-axes for Alaska, Greenland and Svalbard represent in Fig. 2b. Should be made clear.

Response 15: In Fig. 2b, we designed a schematic representation of the topography of

parts of the Arctic and its adjacent regions in two phases; that is, the X-axis has no specific meaning. A similar case can be found in Fig. 2B of Ding et al. (2020 *Science* 369, 578–581) and Fig. 3B of Li et al. (2022 *Proc. Natl. Acad. Sci. USA* 119, e2207199119). To avoid misunderstanding, we placed Fig. 2b in a separate box (see revised Fig. 2).

302-301. Figure 3 needs improving. It is not easy to distinguish clearly some of the curves. Would be helpful to label each curve according to geographical region. Colour alone is not sufficient.

Response 16: Following your suggestion, we have re-compiled Figure 3, in which each curve was labeled according to geographical region and color.

305. Mediterranean region appears to include the Alps and Pyrenees. Were arctic species dispersed from these high mountains, or lowlands as well in this region?

Response 17: Based on the floristic division of Takhtajan (1986 *Floristic regions of the world*. University of California Press, Berkeley, CA), the Mediterranean region indeed includes the Alps and Pyrenees. A lot of biogeographic analyses adopted this division, such as Manafzadeh et al. (2014 *J. Biogeogr.* 41, 366–379), Huang et al. (2018 *Ann. Bot.* 122, 1245–1262), and Gorospe et al. (2020 *J. Biogeogr.* 47, 2442–2456). In the revised manuscript, we added the citation of Takhtajan (1986) in the method part (Lines 228–230).

In this study, we identified three dispersal events from the Mediterranean region into the Arctic, of which one is certainly from high mountains and two are ambiguous. This information is in the legend of figure 3, where this comment is made. So, it is not necessary to further revise the legend of figure 3.

327-328. The number of species removed from each clade should be indicated in Supp. Info. An indication should also be given regarding topology inconsistency across loci and what this might be due to, i.e. incomplete lineage sorting and/or reticulation. Could these causes be distinguished and commented on?

Response 18: It is routine to exclude strongly conflicting taxa in plastid and nuclear trees before combining plastid and nuclear datasets (reviewed by Wang et al. 2014 *Sci. China Life Sci.* 57, 280–286) (see above response 1 to Reviewer 1). In this study, our major objective is to investigate the occurrence variations of the Arctic species through time, not to investigate the evolutionary history of each clade that we selected. Moreover, this information is in Methods, where this comment is made. Thus, here it is not necessary to provide the conflicting causes. In the revised version, we highlighted that we removed the conflicting taxa between the plastid and nuclear trees (Lines 207–209).

A total of 68 species were removed, all of which are non-Arctic species (Lines 210–211). All removed species with an asterisk were listed in revised Supplementary Table 7.

364. Needs explanation. What are the two redefined areas?

Response 19: We have added the explanation of two redefined areas, Arctic and non-Arctic (Line 248), which also occurred in the above part Biogeographic and habitat analyses (Lines 224–228).

Supplementary Information

185. Mandel et al. (2019) is not cited in references.

Response 20: We are very sorry for our negligence. We added Mandel et al. (2019 *Proc. Natl. Acad. Sci. USA* 116, 14083–14088) to the References of Supplementary Information. To make sure that all citations were listed in the References, we have also checked carefully all references in the main text and supplementary information.

References:

- Bennike, O. & Bocher, J. (1990). Forest-tundra neighboring the North Pole: plant and insect remains from the Plio-Pleistocene Kap Kobenhavn formation, North Greenland. *Arctic* 43: 331–338.
- Brochmann C. & Alsos IG. (2021). Origin and dispersal of the North Atlantic vascular plant floras. In Panagiotakopulu E & Sadler JP (eds) *Biogeography in the Sub-Arctic: The past and future of North Atlantic Biota*. John Wiley & Sons Ltd. Pp.85-101.
- Brochmann, C. & Bryusting, A.K. (2008). The Arctic – An evolutionary freezer? *Plant Ecology & Diversity* 1: 181–195.
- Gustafsson ALS, Gussarova G, Borgen L, Ikeda H, Antonelli A, Marie-Orleach L, Rieseberg LH, Brochmann C. 2021. Rapid evolution of postzygotic reproductive isolation is widespread in arctic plant lineages. *Ann Bot.* 129:171-184.
- Hou Y et al. (2016). RAD-seq data point to a northern origin of the arctic–alpine genus *Cassiope* (Ericaceae). *Mol. Phylo. & Evol.* 95:152-160.
- Kadereit, JW. & Abbott R.J. (2021) Plant speciation in the Quaternary. *Plant Ecology & Diversity* 14:105-142.
- Sun, H. (2002). Evolution of arctic-tertiary flora in Himalayan–Hengduan Mountains. *Acta Bot. Yunnanica* 24, 671–688.

Response 21: Thank you very much for these additional references. All related papers have now been cited in the revised version. Compared to the initial submission, we added 11 citations and accordingly listed them in the References. We also deleted one older

reference.

In the main text and supplementary information files, a total of 24 comments were given by Reviewer 1, of which 21 are repeated with the above. We provide a point-by-point response to another three as follows.

151. Stronger? In what way?

Response 22: This sentence has been re-written "....., large ice sheets began to display stronger quasi-100-ka cyclicity" (Line 159).

295. Should this be 'of the Arctic' or 'of parts of the Arctic'?

Response 23: Following your suggestion, we used "of parts of the Arctic" to replace "of the part Arctic" (Line 475).

305. "Mediterranean region" This region appears to include the Alps and Pyrenees. Is this wise?

Response 24: As noted, based on the floristic division of Takhtajan (1986 *Floristic regions of the world*. University of California Press, Berkeley, CA), the Mediterranean region indeed includes the Alps and Pyrenees. A lot of biogeographic analyses adopted this division, such as Manafzadeh et al. (2014 *J. Biogeogr.* 41, 366–379), Huang et al. (2018 *Ann. Bot.* 122, 1245–1262), and Gorospe et al. (2020 *J. Biogeogr.* 47, 2442–2456). In the revised manuscript, we added the citation of Takhtajan (1986) in the method part (Lines 228–230). Please see above Response 17 to Reviewer 1.

Reviewer #2 (Remarks to the Author):

The present manuscript is dedicated to the arctic biome from a botanical-evolutionary perspective. It attempts to shed light on the origin and evolutionary dynamics of the arctic flora by examining different phylogenetic lineages of angiospermous plants. It is based on an elaborate evaluation of the phylogenetic trees of several arctic genera. Included were 32 clades with 3,588 species. This makes the study an important stone for understanding this biome, especially in the light of climate change.

Thank you very much for your positive comments!

The term 'clade' is used in a phylogenetic context. It seems that the authors consistently refer to 'genera' in the taxonomic sense (and not families or orders etc.), so I would recommend using this term.

Response 1: In this study, we selected 32 plant groups, of which 23 are genera (all of which are monophyletic), three are clades within genera, and one is a generic pair. The use of the term "clade" (= monophyletic group) is thus appropriate.

The introductory sections are overall well written and give a brief overview of the geological/climatic/vegetation history of the Arctic. However, most of the references are older than 20 years and more recent studies have not been sufficiently considered. This is also reflected in the statement (lines 54-55): "The evolutionary origin of this unique and fragile biota, however, is poorly understood". This statement was made in 2003! Instead, there has been significant progress on this issue in the last 20 years. Similarly questionable appears the statement (l. 64-65): "Yet, the actual timeframe during which modern arctic flora began to appear remains unknown."

Response 2: Our understanding of evolutionary origins of the Arctic flora has indeed improved in recent years through molecular phylogenetic analyses, but most studies focused on source areas of Arctic species without divergence time estimation and/or model-based ancestral range reconstruction; moreover, evolutionary studies in the Arctic have mostly been limited to single taxa (see above Response 4 to Reviewer 1). In this study, we used a multi-taxon analysis to investigate how the Arctic flora was shaped over time, which is still poorly understood. In the revised manuscript, we first changed "The evolutionary origin of this unique and fragile biota," to "The origin and evolutionary dynamics of this unique and fragile biota," (Line 48), and then deleted the initial Reference 13 and cited Hoffmann & Röser (2009 *New Phytol.* 182, 774–780; Qian et al. 2022 *Global Ecol. Biogeogr.* 31, 396–404) (Line 49). Please see above Response 2 to Reviewer 1.

In the revised version, we first added an introduction of the results of previous molecular phylogenetic studies, "Molecular phylogenetic analyses indicate that the ancestors of some Arctic lineages originated in the mid- to late Miocene. However, these evolutionary studies have mostly been limited to single taxa" (Lines 58–61). The sentence "Yet, the actual timeframe during which modern Arctic flora began to appear remains unknown" was then rephrased to "To determine the actual timeframe during which modern Arctic flora began to appear, molecular phylogenetic study of multiple clades across the angiosperm tree of life is vital" (Lines 61–63). Please see above Response 4 to Reviewer 1.

In some other topics, the current state of knowledge is indeed not correctly reflected either. For example, with regard to the relatively minor importance of in situ speciation in the Arctic (l. 71), which has already been demonstrated in previous studies, e.g. for *Draba*, *Douglasia*, *Saxifraga*, *Cerastium*, etc. In this respect, the present study offers no

new examples or findings at all.

Response 3: We have re-written the part of *in situ* speciation text. First, we added *in situ* speciation in the Arctic and cited three papers (Lines 70–71). Second, the sentence "Moreover, compared to the role of dispersal, *in situ* speciation in the Arctic has long been overlooked as a factor involved in composition of the region" has been rephrased to "Whether both dispersal into the Arctic and Arctic *in situ* speciation have similar trends through time remains unknown, however, due to a lack of concerted analyses of numerous exemplar arctic taxa" (Lines 71–73). Please see above Response 6 to Reviewer 1.

A study comparable to the present work, entitled "Taxon recruitment of the arctic flora: an analysis of phylogenies" (Hoffmann & Röser, *New Phytologist* 182: 774–780), was published in 2009. In this investigation (p. 774) "available molecular phylogenetic studies were evaluated for 148 of 374 genera occurring in the Arctic to determine the relative roles of their independent origins and their diversification in the development of the contemporary arctic flora". Many of the results of this work anticipate supposedly new findings of the present manuscript. However, it is not cited at all in the present manuscript, which is a rather big mistake. The by far much more important role of dispersal compared to *in situ* evolution of arctic species (l. 96) additionally is not new.

Response 4: We sincerely apologize for missing the key paper of Hoffmann & Röser (2009 *New Phytol.* 182, 774–780), which is indeed a foundation of our study. In the revised manuscript, we cited this paper four times (Lines 49, 69, 71, 101), including in discussing the relative role of dispersal compared to *in situ* speciation of Arctic species (Lines 98–101).

The discussion of MDEs identifies "three change points ..., supporting a stepwise model for the formation of the arctic flora" (l. 102). Undoubtedly, there have been many changes over the last 3 million years, including interglacials with less glaciation in the Arctic than today, during which the arctic biome, including its former arctic flora, probably almost disappeared. The space available for arctic flora was certainly severely affected. The role of extinction should be investigated in this context and time frame. The explanation for the drop in the MDE curves at about 0.8 Ma is not convincing (lines 148–152).

Response 5: We entirely agree with you that undoubtedly, extinction can have an impact on the formation of local communities of species. However, to our knowledge, it remains very difficult to analyze extinction dynamics in a multiple-taxon study. Using 32 extant angiosperm clades, our main objective is to determine the initiation age of the Arctic biota and to test whether dispersal and *in situ* speciation through time have similar trends

(Lines 75–77). The similar case can be found in Ding et al. (2020 *Science* 369, 578–581) and Xu et al. (2021 *Nat. Sci. Rev.* 8, nwaaz263), who also only analyzed dispersal and *in situ* speciation dynamics. In this revised version, we added a discussion about the effect of extinction on the formation of the Arctic flora (Lines 160–162).

During the mid-Pleistocene transition (*c.* 1 to 0.8 Ma), climate cooling extreme and an apparent c.100 kyr pacing of glacial and interglacial extremes occurred (Mudelsee et al. 1997 *Earth Planet. Sci. Lett.* 151, 117–123; Diester-Haass et al. 2018 *Earth-Sci. Rev.* 179, 372–391), and large ice sheets began to display stronger quasi-100-ka cyclicality (Miller et al. 2020 *Sci. Adv.* 6, eaaz1346) (Lines 156–159). Undoubtedly, these extreme environmental conditions could have prevented immigrants from occupying the Arctic and might have also promoted increased extinction of Arctic species. Thus, both MDE curves decreased during this period. Nevertheless, severe environmental conditions in the Arctic over the last 3 million years could not lead to the extinction of all lineages; some studies have indicated that some refugia in the Arctic existed during glacial cycles (Bringloe et al. 2020 *Proc. Natl. Acad. Sci. USA* 117, 22590–22596 and Napier et al. 2020 *Ecography* 42, 1056–1067). Moreover, our results indicate that at least seven Arctic endemic species originated before 3 million years (see Supplementary Table 5 for details). In this revised manuscript, the last sentence in the paragraph has been rephrased "These extreme environmental conditions could have prevented immigrants from occupying the Arctic and might have also promoted increased extinction of Arctic species" (Lines 160–162). We hope that you are satisfied with this explanation and revision.

Conclusions such as “dispersal into the Arctic began around 11.8 Ma” (l. 103) and “dispersal into the Arctic began at about 11.8 Ma” (l. 106) also appear questionable, leading to the assumption "that arctic flora had emerged in the early Late Miocene, which is much earlier than the prevailing view that this flora began to appear at about 2–3 Ma”. The most likely explanation is missing from the manuscript. Some phylogenetic lineages with arctic taxa are older than the arctic biome, as noted already in previous studies.

Response 6: Some studies indeed reveal that a few Arctic lineages originated older than the mid-Miocene, but these lineages are not endemic to the Arctic, such as *Pinus pumila* (Eckert & Hall, 2006 *Mol. Phylogenet. Evol.* 40, 166–182), and *Androsace filiformis* and *A. septentrionalis* (Wang et al. 2004 *Acta Phytotaxon. Sin.* 42, 481–499). Thus, we do not know when they occupied the Arctic. In this study, we selected 32 clades and 3,626 species, of which 548 are distributed in the Arctic and 40 are restricted to the Arctic (Lines 80–82). Our sampled taxa can represent the Arctic flora as a whole, as our selected taxa consists of species with various habit and life history traits and belong to 10 orders and 16 families across the angiosperm tree of life (Lines 84–87). We built a time-calibrated phylogeny and performed ancestral range reconstruction for each of these 32

clades. In the revised manuscript, we first excluded wrong species (please see below Response 10 to Reviewer 2), and then re-analyzed our data. Our new meta-analysis indicates that Mediterranean *Artemisia* and western North American *Pleuropogon* dispersed into the Arctic in the early Late Miocene (c. 10.2 Ma, 95% CI: 10.5–9.1 Ma), and during the same period, Arctic *in situ* diversification occurred in three genera (*Artemisia*, *Puccinellia*, and *Ranunculus*) (Lines 108–113). These indeed suggest that some components of modern Arctic flora might have formed in the early Late Miocene. The above-mentioned four genera are not related and distributed in three distantly related families of angiosperms. Thus, we think that it is appropriate to say "Arctic flora might have emerged in the early Late Miocene, which is much earlier than the prevailing view that this flora began to appear at about 2–3 Ma" (in the revised manuscript, we changed "had emerged" to "might have emerged" to tone down our conclusion; Lines 113–115). A similar case can be found in Ding et al. (2020 *Science* 369, 578–581), who said "Our historical reconstruction indicate that an alpine flora had emerged in the THH region by the early Oligocene, mainly by ancestral lineages of Delphineae and Saxifragaceae". Please see above Response 8 to Reviewer 1.

We have had a detailed discussion about why the initiation of the Arctic flora might have occurred in the early Late Miocene: tectonic activities, river erosion, and sea-level drop might have led to the formation of new ecological opportunities for cold-adapted lineages and thereby promoted the initiation of the Arctic flora (Lines 115–124), in agreement with the results of paleovegetational reconstruction, i.e., canopy openness began increasing at 9.7 Ma (Lines 124–125; Fig. 2e).

The conclusion that "most biotic dispersal of the arctic flora has been between the Arctic and western North America, ... whereas there were far fewer dispersal events from other regions" (lines 160-165) should be carefully reviewed. Dispersal along western North America into the Arctic is undoubtedly important and has also been noted in previous studies, but the connections of the vast Eurasian Arctic in particular with southern regions have probably been underestimated in this study. One possible reason is the insufficient consideration of the Eurasian Arctic in this study. As can be seen from the "supplementary information" on the topic "Geographical range and habitat information were mainly compiled from the following sources", no flora at all from the area of the former USSR or the famous "Arkticheskaya Flora" (Tolmachev, A.I. (Ed.), 1960–1987. *Arkticheskaya Flora SSSR*. Vol. 1-10. Moscow, Leningrad) was taken into account, which is actually not possible in this matter. The important relations between the Eurasian Arctic and the South Siberian mountains are not mentioned anywhere in the manuscript. See: Tkach, N.V., Röser, M., Hoffmann, M.H. (2008): Range size variation and diversity distribution in the vascular plant flora of the Eurasian Arctic. *Organisms, Diversity and Evolution* 8: 251-266.

Response 7: The sources listed in “Supplementary Information” for “Geographical range and habitat information” were used to obtain the data for all sampled species, not only for Arctic species, and to perform ancestral state reconstructions for the selected 32 clades. Among the eight sources that we used, Plants of the World Online (<https://powo.science.kew.org/>) and Global Biodiversity Information Facility (<https://www.gbif.org/>) have the distribution information of global species, including those distributed in Russia. Thus, we believe that the information of the sampled species that we obtained should be correct. In this study, we selected 32 angiosperm clades and 3,626 species belonging to 10 orders and 16 families across the angiosperm tree of life (Lines 80–87). Our results indeed indicate that “Most biotic dispersals of the Arctic flora were from western North America (~54%, 14/26 of all dispersal events), whereas there were far fewer dispersal events from other regions (Supplementary Table 6)” (Lines 170–173). This may be because “Arctic tundra covers a bigger area in North America than in Eurasia (Fig. 1), and the Canadian Arctic Archipelago consists of more than 36,500 islands, a feature not seen in Eurasia. These factors suggest that compared to Eurasia, North American Arctic regions can provide more ecological niches for immigrants of the Arctic.” (Lines 175–179). Please see above Response 13 to Reviewer 1.

In this study, our main objective is to investigate “the origin and evolutionary dynamics of the Arctic biota”, not detailed sources of Arctic species, which have been investigated in previous studies (Lines 65–67). In this paragraph, our aim is not to compare dispersal numbers of different regions into the Arctic, instead to highlight conservation priority for the dispersal corridor between the Arctic and western North America owing to more dispersals from western North America into the Arctic (Lines 184–186).

The supplement also states that “Phylogenetic data using DNA sequences were generated in this study or collected from GenBank”. However, a random check of the sequence numbers in GenBank did not reveal any sequence that was generated in the present study. In the case of *Draba*, for example, it was even explicitly stated that the *matK* and *rbcL* sequences had been determined in the context of this study. This was also not true. It would be practical if the new sequences were labelled.

Response 8: In this study, we generated 185 new sequences, which were indeed deposited in GenBank (See details in Supplementary Note). The sequences will all be released upon publication. During this revised process, we labelled new sequences with an asterisk (please see revised Supplementary Table 7). The sentence in Data availability was also rephrased “Newly obtained sequences are available in GenBank under accession numbers OQ617504–OQ617512, OQ623108–OQ623129, OQ703357–OQ703382, OQ703430–OQ703528, and OQ738236–OQ738264” (Lines 275–277). We also deposited our alignments, settings, and timetrees for the 32 clades in Zenodo (<https://doi.org/10.5281/zenodo.7868007>) (Lines 277–279).

I checked only some of the genera studied to see if the arctic species were correctly selected (see Panarctic Flora <http://nhm2.uio.no/paf>, distribution maps in ‘Arkticheskaya Flora USSR’, other sources). I found that several entries were not correct, that some species were not considered as “arctic” or that some arctic species had been overlooked. Endemic arctic species were marked in red, although in many cases these species have a wider distribution range. Unfortunately, the number of such data is quite high, so this should be checked again for all species studied and the analyses should be done again if necessary.

Response 9: This question is almost identical to the one immediately below. Please see below Response 10 to Reviewer 2.

I also checked a few of the genera (“clades”) studied in the manuscript to see if the arctic species were correctly marked as such (see Panarctic Flora <http://nhm2.uio.no/paf>, distribution maps in Arkticheskaya Flora USSR, other sources). I found that several markings on the phylogenetic trees were incorrect: some species were not marked as arctic or that some arctic species were overlooked. Endemic arctic species were marked in red on the trees, although in many cases these species have a wider distribution range. Unfortunately, the number of such data is quite high, so this should be checked again for all species studied and the analyses should be done again if necessary.

Some examples:

- *Artemisia arctica*, *A. tilesii*, *A. leucophylla*, *A. gmelinii*, *A. frigida* are overlooked as arctic species, whereas *A. borealis* is not endemic arctic.
- *Delphinium elatum* is not endemic arctic.
- *Plantago canescens* is not endemic arctic.
- *Plantago elongata* is not arctic.
- *Braya humilis* and *B. glabella* are not endemic arctic.
- *Braya siliquosa* is arctic species.
- *Oxytropis middendorffii* is not endemic arctic.
- *Ranunculus glacialis* is not endemic arctic.
- *Androsace triflora*, *A. chamaejasme*, *A. septentrionalis*, *A. filiformis* are arctic.
- *Papaver lapponicum*, *P. radicum*, *P. mcconnellii* are arctic.
- *Papaver nudicaule* is not endemic arctic.
- *Senecio lugens*, *S. Senecio pseudoarnica* are arctic.

Response 10: Thank you very much for your careful work. We carefully re-checked each species that we sampled one by one, especially Arctic species based on Panarctic Flora (<http://panarcticflora.org/>), Global Biodiversity Information Facility (<https://www.gbif.org/>), and Plants of the World Online (<https://powo.science.kew.org/>).

We made sure that each species used in this study indeed occurs in the Arctic or is endemic to the Arctic.

In our initial submission, the 32 clades that we sampled do not include *Androsace* and *Senecio*, both which were only selected as outgroups. In this revised manuscript, we still excluded *Senecio* owing to sparse taxon sampling, but included *Androsace*, i.e., *Douglasia-Androsace* clade (Supplementary Table 7). First, we confirmed that one species (*Plantago elongate*) that you listed are not Arctic, and also found that another three species (*Androsace lactiflora*, *A. lehmanniana*, and *A. maxima*) are not Arctic. Second, we confirmed that 13 species (*Androsace chamaejasme*, *A. filiformis*, *A. septentrionalis*, *A. triflora*, *Artemisia arctica*, *A. tilesii*, *A. leucophylla*, *A. gmelinii*, *A. frigida*, *Braya siliquosa*, *Papaver lapponicum*, *P. mcconnellii*, and *P. radicum*) that you listed are Arctic. We also found another 22 species (*Artemisia canadensis*, *A. dracuncululus*, *A. jacutica*, *A. sericea*, *A. vulgaris*, *Astragalus tugarinovii*, *Cardamine dentata*, *C. hyperborea*, *C. impatiens*, *Euphrasia hyperborea*, *E. parviflora*, *Festuca rubra*, *Pedicularis albolabiata*, *Plantago eriopoda*, *P. lanceolata*, *Poa annua*, *P. pratensis*, *P. supina*, *Potentilla nivea*, *Rumex confertus*, *R. crispus*, and *Saxifraga hirculus*) that are Arctic. Third, we confirmed that eight species (*Artemisia borealis*, *Delphinium elatum*, *Plantago canescens*, *Braya humilis*, *B. glabella*, *Oxytropis middendorffii*, *Ranunculus glacialis*, *Papaver nudicaule*) that you listed are not endemic to Arctic. In addition, we found another six species (*Carex norvegica*, *Oxytropis sordida*, *Packera hyperborealis*, *Potentilla hyperarctica*, *Saxifraga rivularis*, and *Silene uralensis*) that also occur both in the Arctic and other regions.

In the revised manuscript, a total of 548 Arctic species were finally included, of which 40 are endemic to the Arctic. We re-ran all related analyses and updated the main text, Figs. 2a and 3, and supplementary information (Supplementary Note; Supplementary Figs. 2, 4–7; Supplementary Table 1–3, 5–6). The new analyses identified 131 biogeographic events related to the Arctic species examined here, including 105 dispersal events and 26 *in situ* diversification events (Lines 96–98); both events began at approximately 10–9 Ma, increased slowly at about 7 Ma, increased sharply around 2.6 Ma, and peaked around 1.0–0.7 Ma (revised Fig. 2a) (Lines 27–30). These results are highly congruent with those of our initial submission. That is to say, these new analyses do not change our conclusions, “dispersal into the Arctic and *in situ* diversification within the Arctic have similar trends through time; the initiation and diversification of the Arctic flora might have been jointly driven by progressive landscape and climate changes and sea-level fluctuations since the early Late Miocene”. Thank you again.

Please check the manuscript: Arctic with upper case “A” as region vs. arctic (for example, arctic biome or arctic species) with lower case “a” as adjective.

Response 11: We checked many published papers and found that both "arctic" and "Arctic" can be used as adjective, such as "Arctic flora" in Qian et al. (2022 *Global Ecol. Biogeogr.* 31, 396–404) and "Arctic vegetation" in Pearson et al. (2013 *Nat. Clim. Change* 3, 673–677). In particular, "Arctic" was found as adjective in *Nature* series journals, such as *Nat. Rev. Earth Environ.* (Overeem et al. 2022 3, 225–240), *Nat. Commun.* (Creamean et al. 2022 13, 3537; Liu et al. 2022 13, 3843), and *Nat. Geosci.* (Loranty 2022 15, 515–516). Since the journal that we submitted to is *Nat. Commun.*, we still adopted "Arctic" as adjective in this revised manuscript. I sincerely hope that you agree with us. Many thanks.

Table S8: Reference (72) seems to be wrong, it should be (71).

Response 12: We are so sorry for this mistake. In the revised version, we have carefully checked all references in the main text and supplementary information.

Fig. 3: Explanation of "E" is missing.

Response 13: In the legend of Fig. 3, the explanation of "E" was added.

All in all, a somewhat careless handling of facts can be observed here as well. One cannot avoid the impression that the actual state of current knowledge has not been adequately taken into account in many places in this study and that many things are presented as supposedly new, although they have been known for a long time.

Response 14: We sincerely thank you and Reviewer 1 again for your time and help on our manuscript. We are also so sorry for that we did not acknowledge enough previous studies. In the revised manuscript, we paid special attention to previous studies and added their citations and the related discussion. After excluding wrong species, we re-ran all related analyses. All these revisions and new analyses do not change our conclusions, "dispersal into the Arctic and *in situ* diversification within the Arctic have similar trends through time; the initiation and diversification of the Arctic flora might have been jointly driven by progressive landscape and climate changes and sea-level fluctuations since the early Late Miocene", instead, make these conclusions more robust. We believe that this report is exciting, important and of broad interest to ecologists, evolutionary biologists, paleontologists, physiologists, conservation biologists, and has general appeal to those who enjoy new hypotheses regarding the evolution of Arctic biodiversity. As you said, the study is "an important stone for understanding this biome, especially in the light of climate change". We hope that you will find these revisions adequate and satisfactory.

Thank you for your critical review and comments.
(End of document)

Reviewers' Comments:

Reviewer #1:

Remarks to the Author:

I am mainly happy with the authors' responses to both reviewers' comments and with the new version of the manuscript. I have suggested some minor changes to the text and recommended that some changes should be made to Figures 1 and 3 to improve their visual clarity. These are indicated in the marked-up pdf of the manuscript.

Reviewer #2:

Remarks to the Author:

A brief review of the revised manuscript leads me to the conclusion that essentially stylistic changes were made to the manuscript, but the content objections were only marginally taken into account. Instead, the only response to the objections of the two reviewers is in "Response to Reviewers". Inserting an important study only as an additional number as a citation without discussing this is not a significant improvement of the manuscript. The same can be observed with other suggestions for improvement from reviewers.

Response to the reviewers' comments

Reviewer #1 (Remarks to the Author):

I am mainly happy with the authors' responses to both reviewers' comments and with the new version of the manuscript. I have suggested some minor changes to the text and recommended that some changes should be made to Figures 1 and 3 to improve their visual clarity. These are indicated in the marked-up pdf of the manuscript.

Response: We are pleased to see that you are satisfactory with our revision. Using your new changes and comments, we further made corrections and improvements. First, we accepted all minor edits that you made on the pdf version. Second, we re-compiled Figures 1 and 3. Additionally, we copied your two comments on Figures 1 and 3, and made a point-by-point response to them as follows.

Fig. 1: In this figure ice-sheets are indicated in white according to the key. However, large areas of the sea are also left white which causes confusion. I suggest the sea is coloured blue so that the ice sheets stand out more clearly.

Response: In Fig. 1, the sea was colored in blue. Very good suggestion, many thanks!

Fig. 3: I can barely see the yellow (H) curve in the figure. Could it be made to stand out more using a stronger colour?

Response: In Fig. 3, we have used a stronger color for the H curve, which made it clearer. Many thanks.

Reviewer #2 (Remarks to the Author):

A brief review of the revised manuscript leads me to the conclusion that essentially stylistic changes were made to the manuscript, but the content objections were only marginally taken into account. Instead, the only response to the objections of the two reviewers is in "Response to Reviewers". Inserting an important study only as an additional number as a citation without discussing this is not a significant improvement of the manuscript. The same can be observed with other suggestions for improvement from reviewers.

Response: This is not true. Reviewer 1 is satisfactory with our first revision (please see the above comment of Reviewer 1). In fact, in the first revised manuscript, we have indeed made lots of corrections and improvements based on the two reviewers' comments and edits. **The major revisions that we made in the first revised manuscript are summarized as follows.** First, we carefully re-checked each Arctic

species that we sampled to make sure that the species used in this study are indeed Arctic or endemic to the Arctic. In the first revised manuscript, we included *Androsace*, which was used as the outgroup in our initial submission. We excluded 4 non-Arctic species and 14 species endemic to the Arctic. In addition, we added 35 Arctic species. Accordingly, we re-ran all related analyses and updated the main text, Figs. 2a and 3, and supplementary information. **Second**, We updated the recent progress of molecular phylogenetic studies, changed the description about the current research status, and added a discussion about why both MDE curves decreased around 0.7 Ma and why there appear to be more immigrants from western North America. Compared to the initial submission, we added 11 citations and accordingly listed them in References of the main text. **Third**, we listed all removed species in revised supplementary Table 7 (Supplementary Data 3 in the second version).

In the new revised version, we further improved the manuscript based on the comments and edits of Reviewer 1 and the editor (please see our responses in the Revised author checklist).

(End of the document)